# Uterine Penetrating Wounds in Pregnant Women—Review and Case Study

**DOI:** 10.3390/jcm14030800

**Published:** 2025-01-26

**Authors:** Klaudia Dolińska-Kaczmarek, Bartosz Burchardt, Zuzanna Aurast, Franciszek Ambrosius, Sebastian Szubert, Szymon Rzepczyk, Paweł Świderski, Czesław Żaba

**Affiliations:** 1Forensic Medicine Department, Poznan University of Medical Sciences, ul. Rokietnicka 10, 60-806 Poznan, Poland; zuzanna.aurast@gmail.com (Z.A.); franciszek.ambrosius@gmail.com (F.A.); szymon.rzepczyk@interia.eu (S.R.); pswiderski@ump.edu.pl (P.Ś.); czaba@ump.edu.pl (C.Ż.); 2Division of Gynecological Oncology, Department of Gynecology, Poznan University of Medical Sciences, ul. Polna 33, 60-535 Poznan, Poland; sszubert@ump.edu.pl

**Keywords:** uterine penetrating wounds, pregnancy, emergency medicine, emergency in pregnancy, penetrating wounds, forensic medicine

## Abstract

**Introduction:** Uterine penetrating wounds in pregnant women are rare. Usually, they are caused by traffic accidents, but sometimes they are an effect of violence, accidental injuries, or deliberate self-harm. **Case Report:** We present a case of a stab wound to the uterus in a 25-year-old woman in the 37th week of pregnancy, which was the result of a knife attack. Emergency splenectomy and cesarean section were performed less than an hour after the attack. The child required resuscitation and long-term intensive therapy. Both the woman and the newborn survived; however, the newborn’s condition remained poor. The child died a few months later, according to community-acquired pneumonia. **Discussion:** Cases of wounds penetrating the uterus have been described for many years around the world. They are often the result of violence, most often from the partner. Statistically, pregnant women are more likely to be victims of violence. Sometimes, the wounds penetrating the uterus are accidental and arise, for example, as a result of a fall. A rare but not isolated case is self-mutilation to terminate a pregnancy. Women often inflict such injuries on themselves using sharp tools, sometimes firearms. There are also cases of another person helping to terminate a pregnancy in this way. **Conclusions:** Violence against women, including pregnant women, remains a significant social problem in all regions of the world and poses a challenge to uniformed services, medical professions, and social services. Regardless of the mechanism of injury, in the event of an imminent threat to the life of the mother and fetus, it is crucial to make quick decisions aimed at saving lives.

## 1. Introduction

Pregnant women constitute a small percentage of patients admitted to the Hospital Emergency Department. In the United States, they accounted for 3% of all admissions between 2010 and 2020 and 8.6% of admissions among women aged 15–44. In England and Wales, pregnant women account for about 1.3% of all trauma admissions of women aged 15–46 years [1].

It is estimated that in 1 in 12 pregnancies, an injury occurs, with only about one-quarter of patients hospitalized [2]. According to other available statistics, injuries in pregnancy account for 5–7% of all pregnancies, with 5.4 per 1000 fetal deaths caused by maternal trauma [3]. Injuries during pregnancy increase maternal and fetal mortality. Still, it should be emphasized that in the case of fetuses, gestational age is the strongest predictor, to a lesser extent than the strength or mechanism of injury [2,4]. Traffic accidents causing blunt injuries are considered to be the most common cause of injuries in pregnancy; in the case of penetrating wounds, gunshot and stab wounds are more critical [5]. Statistically, the most commonly reported injuries in pregnancy are fractures, sprains, and dislocations [2,6], while penetrating wounds are an infrequent complication [5]. Regardless of the mechanism of injury, changes in the physiology and anatomy of the mother, as well as exposure to radiation and other possible teratogens, which may be crucial in the diagnosis and treatment process, should be taken into account in managing the pregnant woman [7].

Injuries in pregnant women that affect the fetus, as well as possible complications, have been described in the literature for many years, each time emphasizing the role of traffic accidents in the occurrence of injuries, as well as the importance of gestational age for the survival of the fetus [8]. However, it should not be forgotten that the causes of blunt and penetrating injuries to the uterus in pregnant women also include domestic violence (most often from the partner) and accidental injuries, as well as deliberate self-harm [1,8,9].

The rarity of penetrating injuries means that, regardless of whether they were caused by stab or gunshot wounds, there are few reports on them in the literature, and there are no official statistics. Descriptions are usually based on casuistry [3,5]. The mechanism of their formation depends on the region of the world and the availability of specific tools. In Europe and Asia, sharp-edged tools, such as knives, dominate, while in the United States, gunshot wounds are more common [8,10].

## 2. Case Study

In March 2018, in a Polish village of about 220 inhabitants, a 22-year-old attacker, armed with several knives, broke into the house of his ex-partner, where there were nine people at night. As a result of the attack, two people died, and four were injured. One of the victims was a 25-year-old woman, who was then in the 37th week of her first pregnancy. The patient’s pregnancy was complicated by gestational type I diabetes mellitus and gestational thrombocytopenia. Ultrasound screening tests of the first and second trimester showed no abnormalities.

During the assault, the pregnant woman was repeatedly stabbed in the head, upper limbs, chest, back, and abdomen. During the assault, some family members fled the scene, and concerned neighbors called an ambulance. The patient was admitted to the Hospital Emergency Department in the district hospital in a serious but stable condition, without signs of developed hemorrhagic shock—about 30 min passed from the event to admission to the hospital. During the patient’s physical examination, the heart rate was steady at 90/min, with a pressure of 130/70 mmHg. The abdomen was tense and painful on palpation, peritoneal symptoms were positive, and peristalsis was present. Fetal heart failure was found. Laboratory tests did not reveal any significant abnormalities—due to the short time since the event, the patient did not develop laboratory features of anemia. Due to the condition of the fetus, after an ultrasound examination of the patient, which showed free fluid in the upper left quadrant of the abdominal cavity, the patient qualified for an urgent laparotomy and emergency cesarean section.

At 23 min after admission, less than an hour after the incident, a cesarean section was performed. It is worth noting that providing help in such a short time was undoubtedly an extraordinary success, crucial for further therapeutic effects.

During the exploratory laparotomy performed immediately after the rescue cesarean section, the most severe injuries of the woman were a 4 cm long stab wound in the left lower abdomen, penetrating the uterus through the left round ligament, and a wound in the left mid-abdomen, penetrating the spleen. Inspection of the remaining abdominal organs and placenta did not show any damage. The uterine wound was surgically treated. A splenectomy was performed, and two units of red blood cell concentrate were transfused. A drain was inserted into the peritoneal cavity. The course of the operation and the stay in the surgery ward were uncomplicated; the patient was discharged home on the third day after the surgery in good general condition.

During the emergency cesarian section, a male newborn was extracted, weighing 3080 g. The newborn was extracted in a severe condition, without heart function. The newborn’s condition on the Apgar scale was determined to be 0-0-0-1 at 1, 3, 5, and 10 min, respectively. The newborn was intubated and resuscitated and then transported by neonatological ambulance (N-type ambulance) to a hospital with a higher referral level.

In the examination after transport, the newborn’s condition was described as severe—intubated, artificially ventilated, few own breaths, pupils wide, stiff, non-reactive, increased muscle tone, pale skin, heart rate steady at 120–140/min, regular. The documentation noted the lack of pH from the umbilical cord blood during extraction. Capillary blood gas on admission revealed pH 6.776, BE—28.4 mmol/L, HCO_3_—7.6 mmol/L, and pCO_2_—51.4 mmHg. The wound located on the anterior surface of the chest on the right side was dressed with surgical stitches. In the 24th minute, there was a cardiac arrest—the newborn was resuscitated again. During further diagnostics, an X-ray showed a pneumothorax up to 3 mm. Due to the progressive decrease in red blood cell parameters and active bleeding, on the second day of the stay, another surgical revision of the wound was performed under general anesthesia. During thoracotomy, the damaged blood vessels and a 2-cm hole in the parietal pleura were treated. The wound canal ran obliquely towards the sternum and downwards; the opening in the pleura was located about 7 cm from the entry wound, 2 cm from the sternum, and slightly below. In addition, a cut costal arch was found. In total, 50 mL of blood was sucked out of the pleural cavity, and drainage was left.

The newborn was diagnosed with severe hemorrhagic shock as a result of the wound suffered. In the first blood count, the Hgb level was determined to be 14.3 g/dL (within the normal range); in the following days, a decrease to 6.3 g/dL was observed on the second day, then to 5.1 g/dL on the third day of stay. Intensive treatment was carried out—up to the fifth day of life, low-leucocyte platelet concentrate, fresh-frozen plasma, and one immunoglobulin preparation were transfused five times with good clinical effect and improvement of laboratory parameters. Pressors were used until the seventh day of life. Oxygen therapy was used until the eighth day of life. Until the age of 3 days, the presence of gastrointestinal peristalsis was not found. Total parenteral nutrition was used until the age of 20 days. Enteral feeding was started on the sixth day of life with Nutramigen. The newborn was found to have markers of severe damage to the central nervous system, with features of hypoxic-ischemic encephalopathy. The newborn was discharged in a stable general condition at 43 days of age with a body weight of 3760 g. Clinical improvement was achieved. Diagnoses included severe birth asphyxia, leukomalacia, hypoxic-ischemic encephalopathy, respiratory failure, fetal trauma, hemorrhagic shock, thrombocytopenia, coagulation disorders, and suspected hemangioma in the liver.

In the seventh month of life, the newborn died due to a complication in the form of interstitial pneumonia. The direct cause of death indicated, based on autopsy and histopathology examination was acute respiratory failure in the course of community-acquired pneumonia. The autopsy did not confirm a direct link between the injuries from the first day of life and the cause of death.

## 3. Discussion

### 3.1. Penetrating Wounds to the Uterus in Pregnant Women Around the World

Injuries to the uterus and pelvis in pregnant women may be the cause of abnormal pregnancy, premature placental separation, miscarriage or premature birth, and death of the pregnant woman and fetus [8]. Traffic accidents are the most common cause of their occurrence [10]. The vast majority of injuries in traffic accidents are blunt. Penetrating wounds are most often inflicted with a sharp-edged tool, such as a knife, or with the use of firearms, and the mechanism of their occurrence varies depending on the region of the world and the availability of specific tools [8,9,10].

### 3.2. Europe

In Europe, wounds penetrating the uterus are rare. Still, violence against women, including pregnant women, is not uncommon, with some reports reporting that up to 60% of pregnant women experience violence during pregnancy, with psychological violence being the most common [9,11]. Young, unmarried women with a low level of education are particularly vulnerable [11,12].

There are relatively few reports on the mechanism of uterine injuries in pregnant women in Europe similar to the one described in this paper, particularly concerning fetal damage as a result of direct penetration of the uterus with a knife by another person. One such case is a growing skull fracture in a child described in Italy by Gallo et al.—the woman’s partner wounded her with a knife in the lower abdomen area to harm the fetus. The incident took place at 30 pregnancy weeks. The mother of the fetus was 20 years old at the time. The injuries were not severe, and the woman was treated conservatively—she gave birth to a healthy child naturally at 40 pregnancy weeks. Swelling in the right temporal region was observed in the newborn, which was mistaken for a subcutaneous hematoma. The child was adopted, and at the age of 30 months, he was admitted to a hospital in Verona due to a throbbing lesion on his head—as a result of diagnostics, a growing fracture of the skull, which is a complication of intrauterine trauma, was diagnosed. Surgical treatment was applied well [13]. Intrauterine head injuries caused by a knife can also lead to neurological defects and hydrocephalus, as described by Schultz et al. from Germany—in the incident they described, a woman was injured with a knife at 29 weeks; it was necessary to perform an emergency cesarean section and a neurosurgical procedure [14].

What is particularly noteworthy, especially in the context of violence against pregnant women, is the case of multiple stab wounds to the abdomen and back of a 33-year-old pregnant woman at 36 + 4 weeks of pregnancy, inflicted by her teenage son. The pregnant woman, as in the case we described, was admitted to the Emergency Department conscious, without developed features of hemorrhagic shock. The male newborn was assessed on the Apgar scale at 7 points at 1 min and 8 points at 5 min. The examination showed a wound to the left buttock. At the time of publication of the case, the child’s medical history up to 19 months of age was known—no significant developmental abnormalities were found during periodic examinations [15].

### 3.3. United States of America

Reports of intrauterine injuries of the fetus in the United States were described as early as the 1850s. With the development of the arms industry and easier access to firearms, as well as the advancing automotive sector, the number of reported cases began to increase. In 1913, 44 cases of intrauterine injuries of fetuses as a result of various types of events—including gunshot and stab wounds—were described. In later years, the number of reported cases increased steadily, with lethal injuries to the fetus sustained as a result of wounds penetrating the uterus, usually the result of the actions of third parties (gunshot wounds or stab wounds with knives, pitchforks, sickles, scythes, or animal horns) [8]. Currently, severe injuries in pregnant women are a significant public health challenge in the United States, with the majority of injuries being blunt injuries as a result of traffic accidents (up to about 70%) [3,10]. The second most common cause is falls, especially in the second and third trimester of pregnancy. Penetrating wounds account for only about 1.5% of pregnancy injuries [3].

In the USA, penetrating wounds caused by gunshots are more common than stab wounds, which may be associated with easy and widespread access to weapons. Mangham et al. described the case of a 19-year-old woman in 36 + 6 pregnancy weeks, who was shot several times in the abdomen, buttocks, and thighs. A cesarean section was performed and the newborn was found to have an entrance wound by the medic: 1 cm in the forehead area on the left side and two linear wounds 2 cm long each; within one of them, fragments of brain tissue were visible. CT scans revealed a left-sided subarachnoid hemorrhage, small intracerebral hematomas, and a displaced skull fracture in the parietooccipital region. A neurosurgical procedure was performed. The management of the newborn was similar to the case we described—intensive fluid therapy, antibiotic therapy, and infusion of pressors were initiated. Due to an episode of convulsions, antiepileptic treatment was initiated. During the 12-month follow-up, the child was found to have some neurological deficits and a slight developmental delay [16].

### 3.4. Africa

In African countries, penetrating injuries to the uterus of pregnant women are statistically more common than in Europe or the United States [9]. Cases of fetal puncture wounds were reported there as early as the early 1960s [17]. This condition is due to the greater exposure of women, including pregnant women, to violence. In Egypt, for example, up to 44% of pregnant women experience some form of intimate partner violence, including 16% physical violence and 10% sexual violence [18].

Oumarou et al. from Cameroon described the case of a 28-year-old black patient in the 30th week of pregnancy who suffered a stab wound to the abdomen as a result of slipping and falling on a knife she was holding in her hand. About six hours passed between the event and the patient’s admission to the hospital, and she presented clinical features of hemorrhagic shock (tachypnea, hypotonia, tachycardia). Laboratory tests revealed Hb 9.8 g/dL. As a result of the incident, evisceration and a puncture wound penetrated the uterus, and as a result, a stab wound to the fetal chest. The patient underwent rescue laparotomy and hysterectomy, and the dead male fetus was extracted. On the anterior surface of the fetal chest on the right side, a 2 cm long puncture wound was visible. The patient was discharged from the hospital on the seventh day after the procedure [19].

Non-accidental stab wounds penetrating the uterus were described by Fleming et al. [20]—the victim was a 29-year-old woman in the third trimester of pregnancy. The perpetrator was her husband, who stabbed the woman several times to damage the fetus. When the woman was diagnosed with acute peritonitis, the blood pressure was 80/50 mmHg. Laboratory tests revealed Hb 7 g%. The woman was resuscitated, and then an urgent laparotomy was performed, during which time, the liver and stomach wounds were dressed. In addition, two stab wounds to the uterus were found—an emergency cesarean section was performed, and a female fetus weighing 1400 g was extracted, estimated at 32 pregnancy weeks. The fetus was found to have a through wound in the upper part of the left arm and stab wounds to the chest and thigh. In this case, the child’s Apgar score is not described, but a respiratory failure that developed shortly after birth, exacerbated by haemopneumothorax, is described. The child required intensive treatment in the neonatal ward, where he was additionally diagnosed with radial nerve paralysis. A gradual spontaneous resolution of the paralysis was observed—about 17 weeks after the incident, the child was discharged home. No other neurological problems in the newborn have been reported [20].

Kafih et al. described the case of a 27-year-old woman in the eighth month of pregnancy who was stabbed three times in the area of the right iliac fossa. Emergency laparotomy and cesarean section were performed, during which a dead fetus was extracted, and two wounds to the skull and back were found. The pregnant woman was found to have extensive damage to the uterus and external iliac artery, which was surgically treated [21].

### 3.5. Asia

Asia is also a region of the world where pregnant women are more exposed to violence, in particular in its southern regions. However, traffic accidents are still the leading cause of intrauterine injuries to fetuses [22]. There are also reports of puncture wounds to the uterus, as well as injuries sustained as a result of sexual violence, e.g., rape with a bottle. In the case of stab wounds in Astana (Kazakhstan), two cases were described; in the first one, a pregnant woman suffered injuries to internal organs and the uterus at 26 pregnancy weeks, and the pregnancy was maintained. In the second case, penetrating wounds to the abdomen and pelvis were found, and an emergency cesarean section was performed at 35 pregnancy weeks—the newborn survived [22].

In Russia, the case of a 17-year-old woman in the 33rd week of pregnancy who suffered a stab wound to the uterus as a result of an accidental fall on a kitchen knife was described. The rescue procedure was similar to the case we described. An emergency cesarean section was performed, and the mother’s abdominal wounds were surgically treated. As a result of the incident, the newborn suffered a wound of about 2 cm in the axillary region, which was surgically treated. He also required postoperative resuscitation. Both the mother and the child were discharged from the hospital in good condition without complications [23].

It should be remembered that not only directly inflicted stab or gunshot wounds can lead to injuries resulting in the death of the mother and fetus. Kanawaku et al. described the case of a 32-year-old woman in the seventh month of pregnancy who was admitted to the Hospital Emergency Department after her husband called an ambulance, claiming that the woman fell several times and then lost consciousness. She died a few hours after being admitted to the hospital. The autopsy showed numerous injuries indicating violence against the woman. In the woman, massive bleeding into the abdominal cavity was found due to the detachment of a fibroid as a result of blows with a blunt-edged instrument to the lower abdomen, placental injuries were found—detachment, partial tearing of fragments—which were also associated with numerous blows. Damage to the internal organs of the fetus was also found, which indicates the high strength and intensity of attacks directed at the abdomen [24]. The case is interesting because there are no reports in the literature of blunt trauma directly affecting the internal organs of the fetus in situations other than traffic accidents [10,24].

### 3.6. Violence Against Pregnant Women

Violence against women is still a significant problem in the world. In the European Union, 8% of women have experienced violence [25], and in countries such as India and Bangladesh, it is as much as 29% and 54%, respectively [26]. According to the data, these are most often women aged between 18 and 74, and the violence itself can be psychological, physical, sexual, or economic violence [25]. Even though violence against women is a fairly common phenomenon, even in highly developed countries, penetrating injuries to the uterus in pregnant women in the course of crimes are rare. In the United States, domestic violence or violence by a pregnant woman’s close partner accounts for about 20% of cases of pregnant patients with trauma [4,27,28]. Risk factors for domestic violence incidents include substance abuse by the mother or her immediate environment, unintended pregnancy, previous incidents of domestic violence, and the patient’s marital status [28].

Statistics on violence against pregnant women vary significantly from region to region. It is estimated that between 1.2% and 66% of pregnant women experience some form of violence [9]. The areas with the highest proportion of pregnant women experiencing violence are sub-Saharan Africa, Southern Asia, and Central America [29]. Most often, the perpetrators of violence are known people—particularly partners [9,13,18,29,30].

It should be noted that studies conducted in the United States indicate that pregnant and postpartum women are 16% more likely to be killed than non-pregnant women, and homicide during pregnancy and up to 42 days after childbirth alone is more than twice as high as all other causes of maternal death [31,32]. African-American women, and those who are unmarried and younger than 25 years old, are most at risk, and the most common mechanism of fatal injuries is gunshot. In more than 50% of cases, the perpetrator of the murder is the pregnant woman’s partner [32]. Interestingly, compared to other countries in the world, the United States has the highest risk of homicides associated with pregnancy and early maternity [33].

Violence against pregnant women can have various causes, and in some cases, it is aimed at damaging the fetus. Such a case was described by Gallo et al., in which the partner wounded the pregnant woman with a knife to kill the fetus—the case is discussed in detail in an earlier part of this paper [13]. Similar motifs are also presented by other authors quoted earlier, i.e., Buchsbaum or Fleming et. al. [8,20].

Another cause of violence against pregnant women may be a robbery background, as was the case described by Avensarius et al. [34]—a 19-year-old woman of 29 years of age was stabbed in the abdominal area during a robbery. After 2 h, an emergency cesarean section was performed—the newborn was assessed at 2, 5, and 6 points on the Apgar scale at 1, 5, and 10 min, respectively. The child had respiratory insufficiency and was hemodynamically unstable. A bleeding wound 2 cm long was found, located above the right ear of the newborn. Imaging tests revealed extensive intracerebral bleeding, which was treated neurosurgically. During the neurosurgical procedure, the wound was also repaired. Six months after the event, the child developed left-sided hemiparesis with spastic abduction and external rotation of the left hand [34].

### 3.7. The Issue of Self-Harm as a Method of Abortion

Limited access to legal abortion—regardless of the reason—is conducive to attempts to terminate a pregnancy on one’s own [35]. In some cases, especially in underdeveloped countries with limited access to medical knowledge, even a legal guarantee of legal abortion will not avoid self-abortion attempts, especially among young women [36]. A lack of adequate medical knowledge, limited access to safe medical abortion, and ignorance of the current legal status of a given country can lead to various negative consequences for the health and life of the pregnant woman [35,36,37].

Self-attempts to terminate a pregnancy are not a rare or new topic. Reports of attempts to terminate a pregnancy by self-mutilation with a knife have been appearing in the literature for many years [38]. However, there are rare cases, such as that described by Sakala et al. in 2009 in the USA—an 18-year-old girl was admitted to the Hospital Emergency Department at 24 pregnancy weeks with stab wounds to the abdomen. Initially, she and her partner claimed that she had been assaulted—in the course of the investigation, it turned out that, with her consent, her partner had stabbed her in the abdominal area to terminate the pregnancy. She had attempted to do this but was unable to inflict injuries on herself. The female fetus was extracted dead—the placental vessels were torn, and there was a wound in the umbilical area of the fetus with evisceration. The patient herself returned to total health without complications [39].

There are isolated reports of uterine gunshot wounds aimed at terminating pregnancy. In none of the described cases did the women suffer severe internal injuries, while the newborns did not survive [40,41].

The literature also includes reports of accidental, lethal fetal injuries as a result of self-mutilation—such a case was described by Kaloo et al. in 2000 in Australia—a 34-year-old Aboriginal woman was admitted to the Hospital Emergency Department in the 25th pregnancy week due to a single stab wound to the abdomen, which she inflicted on herself during an alcohol binge. Pregnancy was first diagnosed during the examination at admission. The patient was stable, cardiopulmonary, and fit on admission, and normocytic anemia was observed during the examination. The fetus was immobile, and only cardiac systolic activity was found. Due to the poor prognosis for the fetus, the focus was on protecting the patient. The stillborn fetus was born about eight hours after laparotomy. A stab wound to the anterior surface of the left thigh penetrated the entire thickness of the tissues, and superficial damage to the placenta and umbilical cord was found. Together with the fetus, the patient gave birth to a clot with a volume of about 700 mL. The woman was discharged on the fifth day after laparotomy [42].

### 3.8. Dealing with a Pregnant Woman After an Injury—Clinical Aspects

Pregnant patients after trauma are a heterogeneous group. However, emergency medical procedures implemented for pregnant women who have suffered an injury do not differ significantly from those performed on non-pregnant women. Actions should be adjusted primarily to the clinical condition of the patient and the fetus, with the mother taking precedence. Anatomical and physiological changes in pregnancy, exposure to radiation, and other teratogens should be considered when managing pregnancy, which may be crucial in selecting diagnostic methods and treatment [4]. It is estimated that in the case of injuries sustained in traffic accidents, which are the most common cause of severe injuries in pregnant women [22], about 8% require an emergency cesarean section, with factors statistically increasing the need for it including, among others, severe head injuries and injuries to abdominal organs [43]. Pregnant women who undergo this procedure have a higher risk of complications and death [43].

In severely ill patients, it is recommended to follow the ALS regimen with some modifications. Rescue procedures should be carried out using the ABCDE regimen. However, it should be remembered that the patient should be placed in the left lateral position 15–30° (right side up) or that the uterus should be manually shifted. In the case of cardiac arrest over 20 weeks of pregnancy, perimortem cesarean section is recommended. Defibrillation should be performed similarly to non-pregnant patients, and medication should be administered similarly [4,44].

The data in the literature are divergent for patients who do not require an emergency cesarean section. It is recommended to monitor the fetus using CTG, but there are discrepancies in the duration of monitoring—from every 4 h to even 24 h. There are also discrepancies in the use of anti-D immunoglobulin in women with Rh-D negative factors after injury. Some recommend the administration of immunoglobulin in every eligible patient, while others make its administration dependent on gestational age. There are also inconsistencies in the administration of the tetanus vaccine [45].

## 4. Conclusions

Pregnant women constitute a significant group of trauma patients admitted to Hospital Emergency Departments. The most common causes of injury are traffic accidents, followed by falls, but violence cannot be overlooked as the cause of injury. Violence against women, including pregnant women, remains a significant social problem in all regions of the world and poses a challenge to uniformed services, medical professions, and social services.

Penetrating wounds of the uterus are most often the result of the actions of another person. Still, statistically, the most common perpetrators of crimes against pregnant women and in the short period after childbirth are people close to the woman, particularly partners. Injuries sustained by both the woman and the fetus can pose a real threat to health and life and, in critical cases, lead to death. In the case of the fetus, gestational age is crucial for survival. However, it should not be forgotten that the causes of penetrating injuries to the uterus in pregnant women are also accidental injuries, as well as deliberate self-mutilation, sometimes with the help of another person, that is aimed at terminating the pregnancy.

Regardless of the mechanism of injury, anatomical and physiological changes in pregnancy, exposure to radiation, and other teratogens should be taken into account when managing pregnancy, which may be of crucial importance in selecting diagnostic methods and treatment. In the case of wounds penetrating the uterus, emergency laparotomy is usually necessary to assess and treat the mother’s internal injuries, and often, an emergency cesarean section is also required. In some cases, a vaginal delivery may be performed. It should be borne in mind that the key in such situations is quick action time. If the fetus survives, depending on the injuries suffered, it may be necessary to perform cardiopulmonary resuscitation and intensive care of the newborn, which will depend on the clinical condition and the results of laboratory and imaging tests.

## Data Availability

All data are available in the Forensic Medicine Department, Poznan University of Medical Sciences.

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
