# Peer review of "Uterine Penetrating Wounds in Pregnant Women—Review and Case Study"

_jcm, 2025, doi:10.3390/jcm14030800_

Round 1

Reviewer 1 Report

Comments and Suggestions for Authors

please see attached pdf for revisions

Comments on the Quality of English Language

Comments made for changes, overall thorough review

Author Response

Thank you for the review. 
Our response is in the attachment. 

Best regards, 

Authors 

Reviewer 2 Report

Comments and Suggestions for Authors

Review reports should contain the following:

A brief summary: The objective of this paper is to explore the etiology, treatment approaches, and outcomes in pregnant women presenting with penetrative injuries to the uterus. The primary strength of this paper lies in its focus on a fortunately rare clinical condition, providing valuable insights into a recent case from the authors' practice. This paper could serve as a foundational basis for the development of future guidelines on this topic.

General concept comments
Article: Limitations of the study: It would have been beneficial to include a pediatrician or pediatric surgeon in the author's team, as their expertise could have provided valuable insights into fetal-neonatal injuries, which are often more complex than maternal injuries. Furthermore, a more thorough explanation from the author (or preferably a pediatrician) regarding how the pneumonia in the child from the case was not related to the injuries sustained would have been helpful. I believe the abstract conclusion drawn from this otherwise well-written study is not entirely accurate: "Exposure to radiation and other teratogens should be taken into account when managing pregnancy, which may be of crucial importance in selecting diagnostic methods and treatment." In contrast, all the studies presented, including the author's case, emphasize the need for prompt intervention, as both the mother and fetus were in life-threatening conditions. This highlights that exposure to radiation and other teratogens were not of significant concern in this context. What should be emphasized more is the fact that less than an hour was lost in the treatment of both the mother and child, which is undoubtedly a remarkable success and crucial in cases like this. In addition, there are some language mistakes that should be addressed.

Review: The review thoroughly addressed this highly relevant topic in a well-organized manner, highlighting the existing gaps in knowledge. However, I feel that the conclusion could have been more solid and give better recommendations for clinicians. The references cited are pertinent and contribute to the overall quality of the review.

Specific comments:

Abstract: It may be misleading to state in the abstract that the newborn survived without clarifying that the condition was extremely severe and that the infant passed away seven months later. In addition, I think authors should explore possible recommendations for social and medical surveillance of children who suffered this kind of birth trauma as they might be in a higher risk of neglect or violence.

Introduction: It lacks a little more detail about the penetrative injuries and not focus as much on blunt trauma.

Case Study: Better organization is needed meaning that all the paragraphs about the child should be grouped one after another and all about the mother should be grouped one after another.

Discussion: Line 157 – What is the relevance of the fact that the couple in question was from Colombia?

Conclusion: Line 403 – This is not supported by the data analyzed in the review.

Comments on the Quality of English Language

The English could be improved to more clearly express the research.

Author Response

(The authors gave the same response as above.)
